# Hospitalization Burden of Patients with Kidney Stones and Metabolic Comorbidities in Spain during the Period 2017–2020

**DOI:** 10.3390/metabo13040574

**Published:** 2023-04-18

**Authors:** Javier Sáenz-Medina, Jesús San Román, María Rodríguez-Monsalve, Manuel Durán, Joaquín Carballido, Dolores Prieto, Ángel Gil Miguel

**Affiliations:** 1Department of Urology, Puerta de Hierro-Majadahonda University Hospital, 28222 Madrid, Spain; 2Department of Medical Specialties and Public Health, King Juan Carlos University, 28922 Madrid, Spain; 3Department of Physiology, Pharmacy Faculty, Complutense University, 28040 Madrid, Spain

**Keywords:** urolithiasis, metabolic syndrome, epidemiology

## Abstract

Nephrolithiasis has become an increasing worldwide problem during the last decades. Metabolic syndrome, its components, and related dietary factors have been pointed out as responsible for the increasing incidence. The objective of this study was to evaluate the trends in the hospitalization rates of patients with nephrolithiasis, hospitalization features, costs, and how metabolic syndrome traits influence both the prevalence and complications of lithiasic patients. An observational retrospective study was conducted by analyzing hospitalization records from the minimum basic data set, including all patient hospitalizations in Spain in which nephrolithiasis has been coded as a main diagnosis or as a comorbidity during the period 2017–2020. A total of 106,407 patients were hospitalized and coded for kidney or ureteral lithiasis in this period. The mean age of the patients was 58.28 years (CI95%: 58.18–58.38); 56.8% were male, and the median length of stay was 5.23 days (CI95%: 5.06–5.39). In 56,884 (53.5%) patients, kidney or ureteral lithiasis were coded as the main diagnosis; the rest of the patients were coded mostly as direct complications of kidney or ureteral stones, such as “non-pecified renal colic”, “acute pyelonephritis”, or “tract urinary infection”. The hospitalization rate was 56.7 (CI95%: 56.3–57.01) patients per 100,000 inhabitants, showing neither a significant increasing nor decreasing trend, although it was influenced by the COVID-19 pandemic. The mortality rate was 1.6% (CI95%: 1.5–1.7), which was higher, if lithiasis was coded as a comorbidity (3.4% CI95%: 3.2–3.6). Metabolic syndrome diagnosis component codes increased the association with kidney lithiasis when age was higher, reaching the highest in the eighth decade of life. Age, diabetes, and hypertension or lithiasis coded as a comorbidity were the most common causes associated with the mortality of lithiasic patients. In Spain, the hospitalization rate of kidney lithiasis has remained stable during the period of study. The mortality rate in lithiasic patients is higher in elderly patients, being associated with urinary tract infections. Comorbidity conditions such as diabetes mellitus and hypertension are mortality predictors.

## 1. Introduction

Nephrolithiasis is a high-prevalence disease that can result in a significant morbidity and cost. Increasing rates of urolithiasis in Western countries have been reported since 1970. The overall prevalence nowadays is currently reported to be between 5 and 9% in Europe and 13% in North America [1]. Dietary factors have been suggested as responsible for this trend, such as increased oxalate consumption, increased sodium, or increased animal protein intake [2,3]. Metabolic syndrome and obesity are considered modifiable individual factors through lifestyle changes and have been individually linked to renal lithiasis through epidemiological studies [4,5,6].

Obesity, overweight, and physical inactivity can cause an inflammatory status in different organs and systems, such as the cardiovascular or endocrine system, liver, or kidney. The effect of lipotoxicity on these organs and systems triggers a constellation of disorders called metabolic syndrome. There is a general consensus regarding its main components: obesity, hypertension, hypertriglyceridemia, hypercholesterolemia, or hyperglycemia [5,7,8].

Different epidemiological studies have shown a higher incidence of kidney stone disease in patients with metabolic comorbidities, reporting an odds ratio between 1.25 and 2.2 [6,9]. Along with this, experimental studies have also shown how the systemic inflammatory status present in these kinds of patients triggers an inflammatory response in the kidney cortex and vasculature, leading to kidney injury and a loss of renal function secondary to a higher deposit of crystals and tubular injury [10,11,12].

Metabolic comorbidities also have a negative influence on mortality in patients with urolithiasis. Other causes that could impact mortality rates are severe cardiometabolic dysfunction, coronary heart disease, and sleep apnea [13,14,15].

In Spain, the incidence and prevalence of urolithiasis has mostly been reported at a regional level, although 2 recent studies have reported the global prevalence of urolithiasis to be between 5.06% [16] and 15% [17]. Sanchez Martin’s estimation was based on a review of the literature, while Arias Vega’s was based on a sample population telephone survey. In the former study, the incidence reported was 0.73%, a data point not reported elsewhere. An increase in the prevalence of urolithiasis in Spain, as in other nearby countries such as Italy or Germany, from 1970 to the end of the 20th century has also been documented [2]. Dietary factors have been uniformly pointed out as a possible cause of this increase in Spain [2,16,17]. Moreover, higher rates of urolithiasis have been linked to higher rates of obesity both in adults and children, with the impact of agricultural modernization and the consumption of fast foods or high-fructose diets being cited as causes of this phenomenon [2]. To date, no epidemiological studies have been reported in Spain based on hospitalization burden or official data. Our aim was to analyze the hospitalization burden of kidney and ureteral lithiasis based on the minimum basic database of all the hospitalizations coded for this diagnosis from 2017 to 2020 and to investigate the trends of hospitalization incidence, sex and age distribution, features of hospitalization, costs, and mortality for this cause. The influence of metabolic morbidities on lithiasic hospitality burden and its mortality was also analyzed.

## 2. Methods

### 2.1. Study Design

A retrospective observational study was performed using the minimum basic data set (MBDS) supported by the Health and Social Services Ministry of the Spanish Government. The MBDS is a public registry for both private and public hospitals that includes 98% of hospital admissions and covers 99.5% of the Spanish population. It is published annually and uses the International Classification of Diseases (ICD-10-CM). The Spanish Ministry of Health has validated it with regard to data quality and overall methodology [18].

Hospital admission records of all patients with kidney and/or ureteral lithiasis (ICD-10-CM codes N20.0, N20.1, N20.2 or N13.2 or N20.9) as a principal diagnosis or a comorbidity were obtained between 1 January 2017 and 31 December 2020 in Spain from the MBDS (Table 1). The following variables were also collected: age, gender, other diagnosis (up to 20 additional diagnostic codes), hospitalization length, cost of hospitalization, and outcome (discharge destination). No readmissions were taken into account; we only assessed one admission per patient. The rate of hospitalization was calculated in relation to the official population figure of the National Institute of Statistics, calculated per 100,000 inhabitants [19]. The personal information was delivered to the researchers anonymously, in compliance with the current Spanish and European legislation.

### 2.2. Statistical Analysis

To analyze the burden of lithiasic patients’ hospitalization, the rate of hospitalization was calculated in relation to the official population, and the 95% confidence interval was calculated the exact same way. Univariate analysis was performed with the chi-square test for qualitative variables and with the t Student test for quantitative variables since the age was the unique continuous variable and followed a normal distribution. Multivariate analysis was performed with a back stepwise logistic regression, setting statistically significant differences in both when *p* < 0.05. SPSS 22.0 and Excel were used as statistical programs.

## 3. Results

A total of 106,407 patients were hospitalized with renal and/or ureteralduring the period 2017–2020. The mean age of the patients was 58.28 years (CI95%: 58.18–58.38. IQR = 25), with 60,455 being male (56.8%), 45,595 (43.2%) female, and 7 patients (0.007%) not having their gender recorded. The mean ages of the male (58.72; 95%IC 58.6–58.9. IQR = 24) and female (57.71; 95CI 57.5–57.9. IQR = 27) patients were significantly different (*p* < 0.0001), due to the large size of the population. The median length of stay was 5.23 days (CI95%: 5.06–5.39. IQR = 4) and did not show statistical differences along the years of hospitalization. Nevertheless, the cost of hospitalization showed statistical differences with the cheapest mean cost of hospitalization in 2018 of EUR 3418 per patient (CI95%: 3383–3453) and the highest in 2020 when the mean cost of hospitalization was EUR 3882 (CI95%: 3835–3930), with an overall cost of hospitalization per year of EUR 95,582,479 for kidney and ureteral stones.

### 3.1. Hospitalization Rates for Kidney and Ureteral Stones by Age, Year of Diagnosis, and Recorded Code

The hospitalization rate for kidney and/or ureteral stones as a principal diagnosis in Spain was 30.29 (CI95%: 30.05–30.54) patients per 100,000 inhabitants/year during the period 2017–2020, and the hospitalization rate of patients with lithiasis (principal diagnosis and comorbidity) was 56.7 (CI95%: 56.3–57.01) hospitalizations per 100,000 inhabitants. When stratified by decade of age, those under 20 years old exhibited the lowest rate of hospitalization of 2.85 hospitalizations, as a principal diagnosis, per 100,000 inhabitants (IC 95% 2.62–3.02), and those in their 6th decade exhibited the highest rate of hospitalization of 55.17 hospitalizations per 100,000 inhabitants (CI95%: 54.16–56.19) (Table 2). The annual rate of hospitalization was the lowest in 2020 with 24.89 hospitalizations, probably due to the COVID-19 pandemic, and the highest was in 2019 with 34.46 hospitalizations per 1,000,000 inhabitants (Figure 1). The Spanish population average percent change (APC) of hospitalization for lithiasis from 2017 to 2020 was −3% (CI95%: −13 to 7), which did not reach statistical significance. The mean length of stay stratified by age was stable at between 3.5 and 4.5 days until the 7th decade of life at which point length of stay increased with age, reaching up to almost 9 days in patients older than 90 years old. The estimated average costs per patient were between EUR 2654€ the youngest and EUR 4366 in the oldest patients. Special attention must be paid to the population under 20 years old in which both length of stay and average costs were higher than in the youngest adult population (see Table 2).

### 3.2. Hospitalization Rates for Renal Lithiasis as Main or Secondary Code

Of a total of 106,407 patients, 56,884 (53.5%) were coded as renal or ureteral stone as the main diagnosis of the hospitalization and 49,523 as a secondary diagnosis. Annual hospitalization rate by main diagnosis or comorbidity is showed in (Table 3). The mean age of the patients with a main diagnosis (53.6 years CI95%: 53.5–53.7) was significantly lower than that of those with a secondary diagnosis (63.7 years CI95%: 63.5–63.9), and more women were diagnosed with a secondary diagnosis (45.4 CI95%: 45–45.8) than a main diagnosis (41.2 CI95%: 40.8–41.7). Ureteral stone was the most frequent diagnosis followed by renal stone (see Table 1) when coded as the main diagnosis. When renal or ureteral stone was not coded as the main diagnosis, the most frequent main diagnosis codes were direct complications of the stones, given the fact that renal or ureteral lithiasis were coded as the secondary diagnosis. Therefore, although renal or ureteral diagnoses were not coded as a main diagnosis, most of these patients’ diagnosis should be considered as a direct complication of a renal and/or ureteral stone (Table 4). The main diagnosis coded in young people when renal or ureteral stone was coded as the secondary diagnosis was “non specified renal colic”, with the rate decreasing progressively until the eighth decade and over at which point the main diagnosis coded was “urinary tract infection”, both of which are complications of a renal or ureteral stone (Table 5). Therefore, the older patients coded with a secondary diagnosis mostly had infectious complications of renal or ureteral stones, showing how older patients are more susceptible to developing direct complications.

### 3.3. Mortality and Morbidity Associated with Urinary Stones

A total of 1590 patients (1.5% CI95%; 1.4–1.6) died during hospitalization (Table 6). It is remarkable that, in most of them, lithiasis was diagnosed with a secondary diagnosis, being coded the main diagnoses as infectious complications related to stones (Table 7). Mortality rates increased with age, reaching a maximum in the oldest decades of life (Table 2).

In relation to metabolic comorbidities, lithiasic patients showed a 40.5% (CI: 40.2–40.8) prevalence of metabolic comorbidities, with hypertension (29.3, CI95%: 29–29.6) and diabetes (16.4, CI95%: 16.2–16.6) the most frequently coded, whereas hypercholesterolemia and metabolic syndrome were the least frequent (Table 8). Cumulative risk factors increased with age, especially in the 8th and 9th decades in which more than 20% of the patients presented 2 or more metabolic diseases. Patients over the fifth decade of life showed mostly the presence of at least one metabolic comorbidity, revealing the important prevalence of these complications in lithiasic patients (Figure 2). Univariate analysis performed to explore the influence of the different risk factors on mortality showed age, lithiasis coded as a secondary diagnosis, diabetes, and hypertension as significant mortality risk factors (Table 9). Multivariate analysis, performed using logistic regression, showed that, as found in the bivariate analysis, age as a continuous variable (OR: 1.075; CI95%: 1.07–1.08), lithiasis coded as a secondary diagnosis (OR: 13.04; CI95%: 10.1–16.9), and suffering from diabetes (OR 1.2 CI95%: 1.1–1.3) increased the risk of mortality in hospitalized lithiasic patients, after adjusting for other covariates. Hypertension did not remain as a significant risk factor in the multivariate analysis.

## 4. Discussion

Nephrolithiasis is a high-prevalence disease that results in significant medical expenditures. An increase in the prevalence and incidence of kidney stones has been demonstrated mainly in industrialized countries, whose causes remain unclear [1,2]. Environmental factors such as dietary changes have been historically pointed out as possible causes of these trends. Moreover, high fructose consumption, animal protein intake, and oxalate or sodium consumption have been identified as risk factors for kidney stone formation [20,21].

Nephrolithiasis affects 5–9% of the population in Europe and 13% of North Americans. The overall probability of becoming a stone former varies across the different parts of the world. Stones are slightly more frequent in males than in females and in white Caucasians than in blacks. Kidney stones in the upper urinary tract appear to be related to lifestyle factors, being more frequent in people living in developed countries in which high animal protein consumption or high-fructose diets are more frequent. However, bladder stones are mainly seen in third-world countries and are associated with infections and poor living conditions [22].

In Spain, no official data have been published regarding the epidemiology of urolithiasis, although a recent review of previous Spanish reports has estimated an incidence of 737 cases per 100,000 inhabitants/year and a prevalence of 5066 cases per 100,000 inhabitants [23]. A study on persons aged 40 to 65 using personal telephone surveys shows how, in Spain, renal lithiasis is associated with older age, belonging to higher socioeconomic classes, the existence of family history of urolithiasis, and hypertension, obesity, or overweight [17].

The most important data regarding the hospitality burden of urolithiasis have been developed in the USA in which the rate of hospitalization for upper urinary stones in the year 2000 was 62 cases per 100,000 inhabitants. This rate has decreased by 15% since 1994, although hospital outpatient visits increased by 40% for this cause in the same period of time [24]. In Spain, no data about the rates of hospitalization have been reported until now. Our study has estimated 56.7 patients hospitalized per 100,000 inhabitants. Although the periods of study are different, the rates seem to be comparable.

In France, another study with similar methods has been published, reporting data from 2006 to 2009, although no results were extrapolated to the general population [25]. Nevertheless, the data of the paper can be extrapolated using France’s 2009 general population (64,658,856 inhabitants) from the World Bank (datatopics.worldbank.org, accesed on 17 April 2023). The hospitality burden of lithiasic patients coded with a primary diagnosis would be about 22.3 patients per 100,000 inhabitants, moderately lower than ours, although the periods of time are different.

The admissions rate was the highest in the oldest males both in the USA and Spain (no data was reported in France about gender and age). In Spain, the highest rates of hospitalization correspond to the age range between 50 and 80 years old. In the same way, in patients older than 65 years old, admission rates have been consistently higher than in younger people in the USA. The ratio of male to female has been lower in Spain than in the USA, where it has decreased from 1.86 in 1994 to 1.45 in 2000. However, it would be interesting to know if that decreasing trend was consistent over time in order to know if both ratios were comparable [24].

The mean length of stay is significantly lower in the USA (2.4 days) than in Spain (5.23 days). However, the same trend is consistently observed in both countries in which the length of stay lowers over the years. In the same way, both studies showed an increase in the length of stay as age increases, except for the pediatric population in which it was higher. In Germany, a hospital length of 3.6 days on average has been reported for urolithiasis patients through the German Federal Statistical Office (2020) [26]. Shulz et al., in an analysis based on health insurance claims data, estimated a 6.8 day average length of stay for inpatients with urolithiasis cases [27]. Nevertheless, these data can be inaccurate because, in every center, some therapies can be administered under different circumstances. For example, SWL or ureteroscopy, which are therapies frequently used for kidney or ureteral stones, can be provided in an outpatient or an inpatient hospital setting, depending on hospital resources or physician preferences.

In the USA, an estimated effect on per capita costs of patients with urolithiasis of USD 4472 per year has been reported, being the highest in 45- to 54-year-old patients [24]. An average cost of USD 26,103 for inpatient cases from 2006 to 2009 has also been estimated [28]. In this study, an average cost of EUR 3026 was found. In Spain, most of the patients are treated in the public Spanish health system where costs are reduced.

According to demographic statistics, the proportion of people aged over 65 years or older will rise in the coming years [29]. Because of this, a rise in lithiasic pathology in geriatric patients is expected. In our study, it has been shown how older patients present with more complicated causes of hospitalizations associated with urolithiasis, being more sensitive to present infectious complications. Mortality in geriatric patients is also higher than in younger patients, reaching up to 13% in patients over 89 years old. Arampatzis et al. analyzed current diagnostic and treatment patterns of geriatric urolithiasis patients in emergency departments. It has also been shown how elderly patients have a higher risk of complications and a twofold increased likelihood of being hospitalized [29].

Moreover, it has been reported how mortality increases with age in lithiasic patients [30] and how therapies widely used for the treatment of lithiasis, such as ureteroscopy, almost double mortality in elderly people [31]. In our study, urinary tract infections have been shown to be the most frequent diagnosis coded in elderly patients. It has been widely remarked that urinary tract infections are a strong predictor of mortality and complications for stone surgery [31]. Other factors related to a poor clinical outcome in critical illness, such as vitamin D deficiency, are frequently found in lithiasic patients [32]. Therefore, our findings suggest a higher mortality in elderly patients influenced by a higher prevalence of urinary tract infections.

Furthermore, other comorbidity diseases have been pointed out as predictors of mortality in treated lithiasis patients, such as gender, body mass index, diabetes, cardiovascular diseases, and vitamin deficiency [29,31,32]. In this paper, age, diabetes mellitus, and hypertension have been shown to be mortality predictors both in univariate and multivariate analyses.

Metabolic syndrome is a constellation of disorders, which includes obesity, hypertension, diabetes, and hyperlipidemia [6,9]. In our study, we have shown how metabolic syndrome components become more prevalent as age increases, with hypertension (29.3%) and diabetes (16.4%) being the most frequently coded in lithiasic patients. Epidemiological studies have reported a higher incidence of urolithiasis in diabetic patients with a RR between 1.31 and 1.67 [33]. Furthermore, kidney stone formers exhibit a higher diabetes mellitus incidence than non-stone formers [34].

Hyperglycemia has been related to alterations in renal handling of urinary ions [35] and to increased oxidative stress inducing cellular changes, apoptosis and necrosis, and cellular death, which will develop a stone nidus and a precursor of a kidney stone [36,37].

Concerning cardiovascular morbidities, lithiasic patients develop hypertension with an OR between 1.24 and 1.96, and, conversely, hypertensive patients show a higher risk of developing kidney stones [38,39]. Enhanced generation of endothelial reactive oxygen species in kidney vessels has a key role in the development of endothelial dysfunction associated with low-grade inflammation morbidities such as urolithiasis [40].

The present study has strengths and limitations. The main strengths are the use of the MBDS records which offer practically all hospital admission records from both public and private hospitals, thus increasing the power analysis even of low-prevalence diseases. However, the quality of the MBDS depends on the quality of the discharge report and on the codification process variables. The results obtained in this study provide an accurate assessment of the changes in the national incidence of hospitalizations due to urolithiasis over four years and allow us to estimate the real burden of hospitalization for urolithiasis. The mortality rate by age group provided in this study is useful to know which groups are more likely to develop complications and which risk factors are associated with them. Nevertheless, mortality rate estimations should be interpreted with caution as deaths could occur outside of the hospital environment.

The direct costs to the national health system or private insurance companies, calculated by the MDBS, were calculated using diagnosis-related groups (DRGs) for this disease. Although DRGs have been widely used as a patient classification system for hospital cost analysis, they present the limitation of including an internal variability in comorbid conditions and a lack of sensitivity to different medical practices used in similar diseases.

## 5. Conclusions

Our study provides updated information on the impact of urolithiasis hospitalization and on the costs associated with hospital admissions due to this condition over a four-year period in Spain. The rate of hospitalization for patients with kidney and ureteral stones has remained stable in Spain and is consistent with other international studies. However, the average length of hospital stays is significantly longer, while associated costs are significantly lower.

The mortality rate in lithiasic patients is higher among elderly patients, being associated with urinary tract infections. Comorbidity conditions such as diabetes mellitus and hypertension have been shown to be mortality predictors. Therefore, the utmost attention should be paid to elderly people, especially with diabetes or hypertension.

Future research using the MDBS or other epidemiological approaches can provide more information on urolithiasis and improve the management of this pathology in our country.

## Figures and Tables

**Figure 1 metabolites-13-00574-f001:**
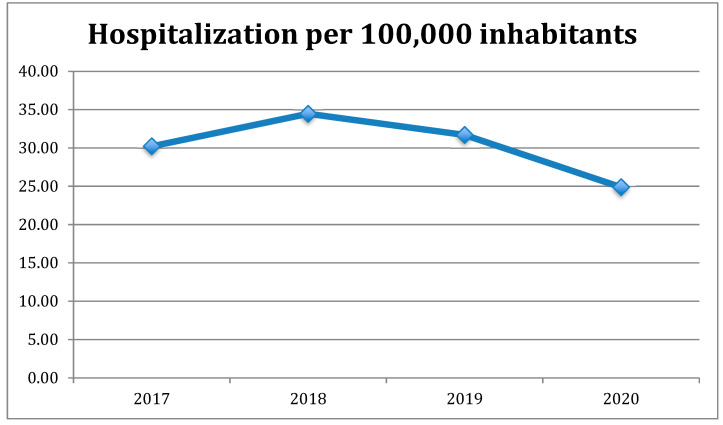
Annual hospitalization for kidney stones in Spain during the period 2017 to 2020.

**Figure 2 metabolites-13-00574-f002:**
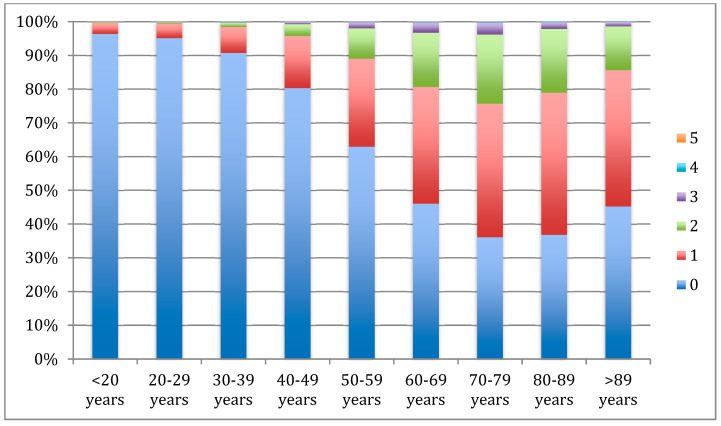
Number of metabolic comorbidities in stone patients by age (%).

**Table 1 metabolites-13-00574-t001:** ICD-10 codes of kidney and ureteral stones when lithiasis was coded as the main diagnosis.

Type of Urolithiasis	Code	Principal Diagnostic	
		N	%
Ureteral stone	N20.1	27,982	49.2
Kidney stone	N20.0	18,052	31.7
Kidney and ureteral stones	N20.2	5865	10.3
Hydronephrosis with renal and ureteral stone obstruction	N13.2	4583	8.1
Nonspecified urinary stone	N20.9	402	0.7
Total		56,884	100

**Table 2 metabolites-13-00574-t002:** Hospitalization rates (rate per 100,000 inhabitants), length of stay (days), mortality (%), and costs (EUR) for kidney stones in Spain (2017–2020) by age.

Age	Rate (95% CI) per 100,000 Inhabitants	Length of Stay x (95% CI)days	Mortality % (95%CI)	Cost x (95% CI) EUR
<20 years	2.9 (2.7–3)	5.1 (4.6–5.56)	0.2 (0.1–0.6)	3076 (2877–3274)
20–29 years	13.6 (13.6–14.9)	3.8 (3.6–4)	0.1 (0.01–0.2)	2654 (2596–2711)
30–39 years	24.9 (24.3–25.6)	4.66 (2.9–6.49)	0.1 (0.05–0.2)	2801 (2753–2848)
40–49 years	38.5 (37.8–39.1)	3.8 (3.7–3.8)	0.2 (0.2–0.3)	3020 (2982–3057)
50–59 years	55.2 (54.2–56.2)	4.4 (4.3–4.5)	0.4 (0.3–0.5)	3404 (3363–3444)
60–69 years	49.1 (48.2–50.1)	5.16 (5.02–5.29)	0.8 (0.7–0.9)	3821 (3770–3871)
70–79 years	42.6 (41.6–43.7)	6.55 (6.3–6.6)	2,1 (1.9–2.3)	4275 (4214–4335)
80–89 years	24.7 (23.7–25.7)	8 (7.8–8.1)	5.8 (5.4–6.3)	4386 (4321–4450)
>89 years	10.4 (9.1–11.9)	8.9(8.5–9.2)	12.8 (11.4–14.2)	4366 (4246–4485)
Total	30.3 (30.1–30.5)	5.2 (5.1–5.4)	1.5 (1.4–1.6)	3593 (3573–3612)

**Table 3 metabolites-13-00574-t003:** Annual hospitalization rate in Spain for kidney stones as the main diagnosis or comorbidity. Annual hospitalization rates per 100,000 inhabitants.

Year	Urolithiasis as Main Diagnosis		Urolithiasis as Comorbidity	
	Rate	95% CI	Rate	95% CI
2017	30.21	29.71–30.71	25.08	24.62–25.54
2018	34.46	33.93–34.99	30.85	30.35–31.36
2019	31.70	31.19–32.21	27.54	27.07–28.02
2020	24.88	24.44–25.34	23.38	22.95–23.82
Total	30.29	30.05–30.54	26.70	26.47–26.94

**Table 4 metabolites-13-00574-t004:** Most frequent causes of hospitalization when a kidney or ureteral stone was not coded as the main diagnosis.

ICD-10 Codes	N	%
Nonspecified renal colic (N23)	3848	7.7
Acute pyelonephritis (N10)	3178	6.3
Tract urinary infection, location nonspecified (N39.0)	2465	4.9
Sepsis, microorganism nonspecified (A41.9)	1314	2.6
Sepsis (Escherichia coli) (A41.51)	1054	2.1
Acute renal failure (N17.9)	878	1.8
Pneumonia (nonspecified microorganism)(J18.9)	682	1.4
Benign prostate hyperplasia (N40.1)	440	0.9
Malignant neoplasm of the bladder (C67.9)	413	0.8
Obstructive and reflux uropathy (N13.9)	400	0.8
TOTAL	14,090	28.8

**Table 5 metabolites-13-00574-t005:** Main cause of hospitalization when urolithiasis was not coded as the main diagnosis by age.

Age	ICD-10 Codes	N	%
<20 years	Nonspecified renal colic (N23)	115	17.7
20–29 years	Nonspecified renal colic (N23)	325	22.7
30–39 years	Nonspecified renal colic (N23)	506	19.4
40–49 years	Nonspecified renal colic (N23)	945	17.1
50–59 years	Nonspecified renal colic (N23)	860	10
60–69 years	Nonspecified renal colic (N23)	575	5.8
70–79 years	Urinary tract infection, location nonspecified (N39.0)	540	5.2
80–89 years	Urinary tract infection, location nonspecified (N39.0)	659	7.7
>89 years	Urinary tract infection, location nonspecified (N39.0)	187	9.4

**Table 6 metabolites-13-00574-t006:** Discharge destinations for patients when lithiasis was coded as main diagnosis or a secondary code.

	Main		Secondary	
	n	%	n	%
Home	55,671	97.9	44,914	90
Other hospital	367	2.2	1079	2.2
Voluntary discharge	134	0.4	179	0.4
Death	64	0.1	1526	3.1
Social residence	31	0.1	619	1.2
Others	593	1	1174	2.2
Unknown	24	0	32	0.1

**Table 7 metabolites-13-00574-t007:** Most frequent main causes of hospitalization in deceased patients when kidney or ureteral lithiasis were coded as main diagnosis or comorbidity.

ICD-10 Codes	N	%
Sepsis, microorganism nonspecified (A41.9)	163	10
Tract urinary infection, location nonspecified (N39.0)	49	3.1
Pneumonia (nonspecified microorganism)(J18.9)	45	2.8
Sepsis (Escherichia coli) (A41.51)	42	2.6
Viral pneumonia (J12.89)	42	2.6
Acute renal failure (N17.9)	41	2.6
TOTAL	382	23.7

**Table 8 metabolites-13-00574-t008:** Metabolic syndrome component frequency (95% CI) in all lithiasic patients.

Age	Total	E88.81	E10/11	I10	E66/Z68	E78.0	E78.1
<20 years	1680	0(0)	17(1)	12(0.7)	37(2.2)	2(0.1)	0(0)
20–29 years	4050	0(0)	44(1.1)	53(1.3)	109(2.7)	10(0.2)	4(0.1)
30–39 years	9469	3(0)	174(1.8)	411(4.3)	370(3.9)	56(0.6)	29(0.3)
40–49 years	17,391	21(0.1)	784(4.5)	2231(12.8)	935(5.4)	209(1.2)	137(0.8)
50–59 years	22,906	34(0.1)	2798(12.2)	6200(27)	1616(7)	662(2.9)	192(0.8)
60–69 years	21,183	41(0.2)	4823(22.7)	8592(41.5)	1755(8.3)	921(4.3)	187(0.9)
70–79 years	16,585	21(0.1)	4958(29.9)	7983(48.1)	1421(8.6)	821(4.9)	129(0.8)
80–89 years	10,814	8(0.1)	3291(30.4)	4869(45)	742(6.9)	399(3.7)	56(0.5)
>89 years	2191	2(0.1)	559(25.5)	840(38.3)	70(3.2)	70(3.2)	7(0.3)
Total	106,277	130(0.1)	17,448(16.4)	31,191(29.3)	7055(6.6)	3150(3)	741(0.7)

E88.81: metabolic syndrome; E10/11: diabetes; I10: hypertension; E66/Z68: overweight; E78.0: hypercholesterolemia; E78.1: hypertriglyceridemia.

**Table 9 metabolites-13-00574-t009:** Association of demographical and metabolic syndrome traits and risk of mortality in patients with renal lithiasis using a univariate analysis (categorical variables: chi^2^, continuous variables: *t* test). Results are given with 95% confidence interval in brackets. ns: *p* > 0.05.

	Odds Ratio	Mean Difference	*p*
Age (years)		20.35 (19.5–21.2)	<0.0001
Gender (Male/Female)	1.01 (0.9–1.1)		ns
Lithiasis coded as comorbidity	28.2 (22–36.3)		<0.0001
Metabolic syndrome (E88.81)	0.51 (0.07–3.7)		ns
Diabetes (E10/11)	2.3 (2.1–2.5)		<0.0001
Hypertension (I10)	1.24 (1.1–1.4)		<0.0001
Overweight (E66/Z68)	1.15 (0.96–1.4)		ns (0.08)
Hypercholesterolemia (E78.0)	1.17 (0.9–1.5)		ns
Hypertriglyceridemia (E78.1)	1.2 (0.7–2)		ns

## Data Availability

The data sets generated and/or analyzed during the current study are not publicly available due to privacy or ethical restrictions and can be available from the corresponding author on reasonable request.

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
