# Peer review of "Hospitalization Burden of Patients with Kidney Stones and Metabolic Comorbidities in Spain during the Period 2017–2020"

_metabolites, 2023, doi:10.3390/metabo13040574_

Round 1

Reviewer 1 Report

Mean age of the patients was 58.28 years (CI 95% 58.18-58.38); 56,8% 22 were male and the median length of stay was 5.23 days (CI95% 5.06-5.39).

Comment: The range 58.18-58.38 may be correct but does not give any information regarding the range of ages from <20 to >89 years. Suggest using another statistic such as IQRs. Likewise for length of stay and other statistical values given if appropriate.

 Several factors have been suggested as responsible for this trend, such as 43 dietary factors or global warming [2].

Comment: How could global warming affect the change over a short time?

From a search of Google Scholar:

Dyslipidemia and kidney stone risk

FCM Torricelli, SK De, S Gebreselassie, I Li… - The Journal of …, 2014 - Elsevier

… The origin of kidney stones is multifactorial with age, gender, race/ethnicity, climate and
occupation described as 
risk factors.1, 2, 3 Metabolic syndrome is also currently associated with …

ave ite Cited by 83

Obesity and kidney stone disease: a systematic review.

A Carbone, Y Al Salhi, A Tasca, G Palleschi… - Minerva urologica e …, 2018 - europepmc.org

… Epidemiologic studies have demonstrated that the stone risk … renal stones disease incidence.
The aim of this systematic review was to investigate the prevalence, morbidity, 
risk factors …

ave ite Cited by 116

Nutrition and kidney stone disease

R Siener - Nutrients, 2021 - mdpi.com

… risk factors for kidney stone formation. Specific nutritional therapy, based on dietary
assessment and metabolic evaluation, has been demonstrated to be more effective than general …

ave ite Cited by 52

Dietary and lifestyle risk factors associated with incident kidney stones in men and women

PM Ferraro, EN Taylor, G Gambaro, GC Curhan - The Journal of urology, 2017 - Elsevier

… proportion of incident kidney stones that could be attributable to these modifiable risk factors
… and NNTP for several modifiable 
risk factors for kidney stones and combinations in 3 large …

ave ite Cited by 162 Related articles All 8 versions

Metabolic risk factors in children with kidney stone disease: an update

FR Spivacow, EE Del Valle, JA Boailchuk… - Pediatric …, 2020 - Springer

… We conclude that specific urinary metabolic risk factors can be found in most children with
kidney stones, with hypercalciuria and hypocitraturia being the most common diagnoses. …

ave ite Cited by 15

Table 7. Most frequent main causes of hospitalization in deceased patients when kidney or ureteral 180 lithiasis were coded as main diagnosis or comorbidity. 181 ICD-10 codes

N

%

Sepsis, microorganism non specified (A41.9)

163

10

Tract urinary infection, location non specified (N39.0)

49

3.1

Pneumonia (non specified microorganism)(J18.9)

45

2.8

Sepsis (Escherichia coli) (A41.51)

42

2.6

Viral pneumonia (J12.89)

42

2.6

Acute renal failure (N17.9)

41

2.6

TOTAL

382

23.7

Comment: Infection risk increases when serum 25-hydroxyvitamin D concentrations are low. Suggest that the journal literature be searched regarding the role of vitamin D in reducing risk of the infectious diseases listed above.

An update of the effects of vitamins D and C in critical illness

…, C Starchl, E Dresen, C Stoppe, K Amrein - Frontiers in Medicine, 2023 - frontiersin.org

… with severe vitamin D deficiency (<12 ng/ml), 28-day mortality was significantly lower in
the … the initial phase of major 
surgery that reliably detects vitamin D deficiency (<20 ng/ml). …

Table 8. What are the numbers in ()?

Significant digits. The general rule is that no more non-zero digits should be given than are justified by the uncertainty of the value.

See "Too many digits: the presentation of numerical data"

https://www.ncbi.nlm.nih.gov/pmc/articles/PMC4483789/

If the uncertainty is greater than about 7%, only two non-zero digits are justified.

P values should be given to two decimal places unless the first two are 00 or the number lies between 0.045 and 0.054. If the first two are 00, then only one non-zero digit can be given.

Thus,

cost hospitalization in 2018 of 3418.49 euros per patient (CI95%: 3383.78-3453.20) to the 119 highest in 2020 when the mean cost of hospitalization was 3882.71 euros (CI95% 3834.97-120 3930.46), being overall cost of hospitalization per year of 95.582.479 euros for kidney and 121 ureteral stones.

Would be better if costs were given in only three significant digits, e.g., 3880 euros

was -3.26% (CI95%: -12.55-7.01),

Comment: Probably should be

was -3% (CI95%: -13 to 7),

20-29 years

13.55 (13.04-14.88)

3.81 (3.62-4)

0.1 (0.01-0.2)

2654 (2596-2711)

Better as

20-29 years

13.6 (13.0-14.9)

3.8 (3.6-4.0)

0.1 (0.01-0.2)

2650 (2600-2710)

Please review all numbers in abstract, text, tables, and figures and adjust accordingly.

The manuscript has many simple grammatical errors that should be corrected by a medical text editor well-versed in English.

Reviewer 2 Report

The manuscript summarizes the analysis of a large population of over 100000 patients hospitalized with renal and/or ureteral lithiasis. The analysis includes the age structure of hospitalized patients, hospitalization rates for renal lithiasis as main or secondary code, the most frequent causes of hospitalization when kidney or ureteral stone was not coded as 168
the main diagnosis, discharge destinations for patients when lithiasis, most frequent main causes of hospitalization in deceased patients when kidney or ureteral lithiasis were coded as the main diagnosis or comorbidity, the frequency of metabolic syndrome components in lithiasic patients, and association of demographical and metabolic syndrome traits and risk of mortality in patients with renal lithiasis.

It is a solid, well-done and well-described study. I wonder it fits the scope of the journal; comorbidity with metabolic syndrome is only one of the parameters analyzed and it seems to be the only reason for submission to the journal. In my opinion, it would be more appropriate for a nephrology/medical epidemiological journal. However, it may be marginally fitted, after all, the formation of kidney stones is a result of altered metabolism. The demonstration of the age dependence of comorbidity with components of metabolic syndrome on the large population is interesting.

Lines 141: What is the assumed criterion for the classification of patients as pediatric? In my country it is <18 years.

How the decrease in  the comorbidity of lithiasis with metabolic syndrome in the oldest age group be explained?
